# Custom-Made 3D-Printed Implants for Anterior Column Reconstruction in the Upper Cervical Spine after Intralesional Extracapsular Excision—Report of 2 Cases and Literature Review

**DOI:** 10.3390/jcm11206058

**Published:** 2022-10-13

**Authors:** Marco Girolami, Cristiana Griffoni, Emanuela Asunis, Luigi Falzetti, Stefano Bandiera, Giovanni Barbanti Brodano, Riccardo Ghermandi, Valerio Pipola, Silvia Terzi, Eleonora Pesce, Donato Monopoli Forleo, Marco Cianchetti, Maria Rosaria Fiore, Livio Presutti, Milena Fini, Alessandro Gasbarrini

**Affiliations:** 1Department of Spine Surgery, IRCCS Istituto Ortopedico Rizzoli, 40136 Bologna, Italy; 2Instituto Tecnológico de Canarias (ITC), 35003 Las Palmas Gran Canaria, Spain; 3Proton Therapy Unit, Hospital of Trento, Azienda Provinciale per i Servizi Sanitari (APSS), 38122 Trento, Italy; 4Radiotherapy Unit, National Center of Oncological Hadrontherapy (CNAO), 27100 Pavia, Italy; 5Otolaryngology and Audiology Unit, IRCCS Azienda Ospedaliero-Universitaria Policlinico di Sant’Orsola, 40138 Bologna, Italy; 6Department of Experimental, Diagnostic and Specialty Medicine—DIMES, Alma Mater Studiorum University, 40126 Bologna, Italy; 7Scientific Direction, IRCCS Istituto Ortopedico Rizzoli, 40136 Bologna, Italy

**Keywords:** 3D-printing, anterior reconstruction, upper cervical region

## Abstract

The use of three-dimensional (3D)-printed custom-made implants is spreading in the orthopedics field for the reconstruction of bone losses or for joint replacement, thanks to their unparalleled versatility. In particular, this novel technology opens new perspectives to formulate custom-made fixation strategies for the upper cervical region, sacrum and pelvis, where reconstruction is challenging. We report and analyze the literature concerning upper cervical reconstruction with 3D-printed personalized implants after tumor surgery, and discuss two cases of patients where this technology was used to reconstruct the anterior column after extracapsular debulking of C2 recurrent chordoma at our institution.

## 1. Introduction

In the past three decades, there have been tremendous technical and technological advances that have profoundly changed spine surgery, allowing what seemed to be impossible before. The spread of rigid segmental fixation paved the way for more complex reconstructions, thus, allowing more aggressive approaches in the treatment of primary bone tumors. Given the encouraging results achieved in terms of local control and overall survival, increasing interest has grown towards more refined reconstructions of the spine.

The advent of three-dimensional (3D) printing techniques, also known as additive manufacturing, with their unparalleled versatility, have offered a very attractive prospective in the reconstruction of substance losses, such as those produced by the resection of musculoskeletal neoplasms [1].

Three-dimensional printing refers to the process of fabricating a physical model through successive layering of powder-like materials (including Ti6Al4V, cobalt-chromium alloy, and stainless steel) based on a volumetric digital image generated by computer-aided design (CAD). It can fabricate an implant tailored to the specific anatomy of the individual patient in a controllable manner to enhance the primary immediate postoperative stability. It can also produce size-controllable micropore structures, which can lower the elastic modulus of the metals, decrease the stress at the solid parts of the implant, and promote integration between metal and bone at the contact surface.

We recently reported a prospective observational study on custom-made 3D-printed titanium reconstruction of vertebral bodies performed in 13 patients who underwent en bloc resection for primary spinal tumors [2]. The results from this series suggested that 3D printing can be effectively used to produce custom-made prosthesis for anterior column reconstruction.

In particular, the upper cervical region has a unique biomechanical function that makes reconstruction in this region challenging. Instrumentation-related complications, such as construct subsidence, migration, and pseudoarthrosis are common due to the lack of an optimal implant. Thus, the novel 3D technology is helpful for spine surgeons to perform safer and more adequately planned cervical surgeries.

We present here two cases of patients affected by C2 chordoma treated by means of extracapsular debulking and reconstruction of the spine with customized 3D-printed prosthesis.

A chordoma is a low-grade, slow-growing but locally invasive and locally aggressive tumor that belongs to the sarcoma family of tumors. Chordomas arise from remnants of the notochord and occur along the midline spinal axis between the clivus and the sacrum, anterior to the spinal cord. The location distribution of chordomas is 50% sacral, 35% skull base, and 15% occur in the vertebral bodies of the mobile spine (most commonly the C2 vertebrae followed by the lumbar then thoracic spine). Overall 5-year survival is approximately 50%, and treatment is en bloc surgical resection (if technically feasible) followed by high-dose conformal radiation therapy such as proton beam radiation [3].

## 2. Materials and Methods

In 2017 and 2018, two patients with C2 chordoma (Enneking stage IB), underwent extracapsular debulking at our institution, with upper cervical reconstruction using customized 3D-printed vertebral bodies. Demographic data of the two patients are reported in Table 1.

### Design of the Implant

Prior to surgery, the design of the prosthesis started from a preoperative thin-cut (1–1.5 mm) computed tomography (CT) that allowed evaluation of the patient-specific anatomy such as shape, width, and length of the endplates, to the extent of the planned resection. Based on these data, a CAD model of the spine was generated, and a virtual implant was designed.

The virtual model was visualized prior to production, to allow further refinements (i.e., fixation technique) until the production of the final version of the prosthesis. Finally, the approved model was fabricated by successive layering of melted Ti_6_Al_4_V powder (Arcam AB, Mölndal, Sweden). The design and the fabrication of the implant were performed at Instituto Tecnològico de Canarias, Las Palmas, Spain, in a time span that was kept within 2 weeks from initial consultation with the senior author (AG). The images showing this procedure are reported as Appendix A.

## 3. Results

### 3.1. Case Report 1

The first case that we report, is of a 65-year-old man in whom a local recurrence of chordoma was detected at the site of an extracapsular debulking (and reconstruction with allograft stabilized with a carbon plate) at C2 that he underwent 2 years before. That first surgical treatment had been performed due to local progression after heavy-particles radiation therapy (70.4 Gy/16 ft. of carbon-ion) that the patient underwent 2 years before that surgery. MRI showed (Figure 1) recurrence of the tumor in the site of the previous surgery, surrounding the fibular graft and the plate (Figure 2), extending into the epidural space (Bilski grade 1C). Vertebral arteries (layer F) were not involved by the tumor as well as the posterior elements.

Despite these findings, the patient was complaining of only mild neck pain and was neurologically intact. Revision surgery consisted of hardware (and graft) removal and extracapsular tumor debulking through a single anterior submandibular extraoral-retropharyngeal pre-vascular approach [4]. A custom-made titanium 3D-printed prosthesis was used for reconstruction (Figure 3) (see Section 3.3 for details). Post-operative course was uneventful, and the patient was able to return to work (dentist) after 4 weeks. Pathological examination of the surgical specimen confirmed the diagnosis of CHO without evidence of de-differentiation.

At 3, 6, 9 and 12 months follow-up, the implant showed good stability suggesting firm osteointegration (Figure 4). At 18 months follow up, a second recurrence of tumor was detected for which another revision surgery was performed for tumor debulking and spinal cord decompression through a double (anterior and posterior) approach (Figure 5). This allowed exploration of the implant–bone interface and direct visualization from which strong osteointegration could be confirmed. The patient experienced a severe neurological complication (ischemic brain injury) due to a major intraoperative vascular injury of carotid artery. The patient died after 2 months for the sequelae of the surgery.

### 3.2. Case Report 2

The second case that we report, is of a 75-year-old man in whom a recurrent C2 chordoma progressed locally despite heavy-particles radiation therapy (74 Gy/37 ft. of proton-ion). The patient had already been submitted to extracapsular debulking (without reconstruction), after frozen-section diagnosis of chordoma, 6 months before the radiation therapy. The CT scan showed pathological fracture due to progression of disease (Figure 6).

MRI showed the lesion replacing the vertebral body and the *dens*, extending into the posterior elements via the pedicles and extracompartimentally both into the epidural space causing spinal cord compression (Bilski grade 2) and in the prevertebral (layer A) space. Vertebral arteries (layer F) were not involved by the tumor (Figure 7).

Despite these findings, physical examination showed only chronic sequelae (mild swallowing deficit, right facial nerve palsy and mild ataxia) of a previous right cerebellar infarction that the patient suffered from 20 years before.

Taking into account the involvement of both the anterior and posterior columns, revision surgery was planned to be performed through a double (anterior and posterior) approach. The first stage consisted of tumor debulking, ventral decompression of the spinal cord and anterior column reconstruction with a custom-made titanium 3D-printed prosthesis (see Section 3.3 for details) thorough an anterior submandibular extraoral-retropharyngeal pre-vascular approach [4]. The second stage consisted of posterior decompression, debulking of the posterior extension of the tumor and occiput to C4 instrumented fusion with occipital plate and lateral mass screws (Figure 8). Post-operative course was uneventful.

At 6 and 12 month follow-ups, CT-scan and X-ray showed good stability of the implant from which osteointegration can be assumed. The patient died 32 months after the surgery.

### 3.3. Reconstruction

In both cases, after thorough extracapsular tumour debulking, reconstruction was performed with a custom-made 3D-printed titanium implant.

The prosthesis was inserted accommodating the anterior arch of C1 into a saddle (proximally) and with progressively increasing length until slight distraction of the anterior column could be felt, indicative of a good fitting. Distally, the base of the implant was laid on the upper endplate of the adjacent vertebral body (C3) to which it was secured with two convergent screws through an integrated anterior plate.

To ensure proper fitting of the prosthesis, a series of three implants was produced for each case: one of the expected length (measured on the preoperative CT scan), one shorter by 2 mm, and one longer by 2 mm.

## 4. Discussion

In this study, we assessed the effectiveness and safety of spinal reconstruction using customized 3D-printed patient-specific implants in the upper cervical region in two patients affected by C2 chordoma. It might appear to be misleading to compare the present report of the treatment of chordoma, with others that include other primary bone tumors of spinal metastases. However, because of the complex regional anatomy of the high cervical spine (which includes important and functionally relevant structures such as vertebral arteries, esophagus, trachea, spinal cord and cervical spinal nerves) en-bloc resection with wide/marginal *tumor-free* margins (which would be the recommended treatment for malignant primary bone tumors, such as chordoma or chondrosarcoma) is technically unfeasible in most cases. Therefore, intralesional extracapsular (or gross total) excision remains the only reasonable option, thus, bridging the gap in the surgical treatment of such different entities. These limitations to fulfill oncologic appropriateness, along with the difficult exposure (that makes incomplete excisions more likely) explains why oncological outcomes (local control and overall survival) tend to be poorer for primary bone tumors arising in this region. Since oncologic outcomes of the treatment of chordoma goes beyond the scope of the present report, they are briefly summarized in Table 1.

Anterior column reconstruction at this level is as well limited by the same constraints, especially, achievement of a reliable and firm proximal fixation might be hard to obtain, thus, making additional posterior stabilization indispensable. Several techniques have been described such as fashioning of the proximal end of a bone graft (auto- or allogenic) into the shape of a clothespin, along with a proper length selection, as the most classical way to achieve primary stability [4]. In this way a saddle is created into which the anterior arch of C1 can be accommodated.

In case reconstruction is performed with a mesh cage, it can be opened and shaped as a “T” allowing an analogous saddle mechanism and, at the same time, creating a plate through which screws can be inserted (into the anterior arch or directed to the lateral masses) to provide more rigid fixation [5,6].

The same fixation strategy can be used when a plate is used with a neutralizing principle, but it can be limited if C1 has to be excised as well, since the thickness of the clivus is such that only very short screws can be inserted [7].

Alternatively, plates can be used with a buttress principle and fixated only distally to limit migration of the graft/mesh cage.

Anecdotal use of full thickness osteo-cutaneous flaps has been reported, particularly after radiotherapy [8].

Three-dimensional printing technology opened a crack to the development of custom-made fixation strategies, this being particularly intriguing in the upper cervical spine, as in the sacrum and pelvis.

In both the reported cases, a titanium prosthesis was taken into consideration for the reconstruction (allowing a tailored solution for proximal fixation), since further radiation was prevented by having already reached spinal cord tolerance.

A total of 26 cases of cervical spine reconstruction using personalized 3D-printed prosthesis after debulking (for both primary and metastatic tumors) were reported in 11 papers listed in the PubMed database (search: “cervical C2 tumor reconstruction”) between 1 January 2016 and 30 August 2022 (Table 2). The most common proximal fixation strategy reported was by including trajectories for divergent screws in the lateral masses of C1 within the implant (four reports), or through integrated plates (one report), while only one reported the use of a “saddle”. In those cases where C1 had been excised, completely or partially, because infiltrated by the tumor, primary stability was achieved using the occipital condyles as a proximal point of fixation (two reports) or, through an integrated plate, on the clivus (one report). For reconstructions below C2, the reported strategies for fixation became the same as those used distally: at least two screws into the vertebral body through the implants, or through integrated plates (one report each).

In addition to these macroscopic details, it must be considered that the progressive layering process allows production of a lattice structure of such regularity and with predetermined surface characteristics that the potential for bone ingrowth is high, leaving only a minimal percentage (up to <10%) of the actual volume of the implant being occupied by titanium.

All the authors described combined (anterior–posterior, or posterior–anterior) approaches, most frequently staging the procedures. Surgical planning is primarily to be taken depending to the local extension of the disease, but additional posterior approach might be not necessary for stability if a 3D-printed implant is used. Of the reported cases, one underwent excision and reconstruction through a single anterior approach and did not experience any mechanical consequence.

Mismatch between the size of the implant and that of the gap produced by the resection might be an unsolvable problem with 3D-printed implants. To overcome this potential problem we produced three differently sized implants of 2 mm height increase. Other authors [10,13,14,16,17] were reported to have produced more implants of different sizes ranging from 2 [14] to 32 [10]. This needs to be taken into consideration in an eventual cost-effectiveness analysis on the use of 3D-printed implants [20,21].

The role of 3D-printing is growing both in orthopedics and neurosurgery. Among the benefits of 3D-printing, it must be mentioned that it requires detailed planning preoperatively and commits surgeons to its meticulous respect. Furthermore, this same technology can be used to produce models that can be used for patient counselling, or training for follow up, or even experienced surgeons [22,23,24].

## 5. Conclusions

Location of spinal tumors in the upper cervical spine provides challenges both for tumor excision, since satisfactory exposure is difficult to achieve (often requiring multiple approaches), and reconstruction, especially for proximal fixation. Three dimensional printing allowed the reported authors to design patient-specific solutions, not only for shape and size of the implants, but also for the method to achieve proximal fixation. Despite this, in cases of tumor extension to posterior elements, anterior fixation alone is limited, and posterior stabilization is generally required.

## Figures and Tables

**Figure 1 jcm-11-06058-f001:**
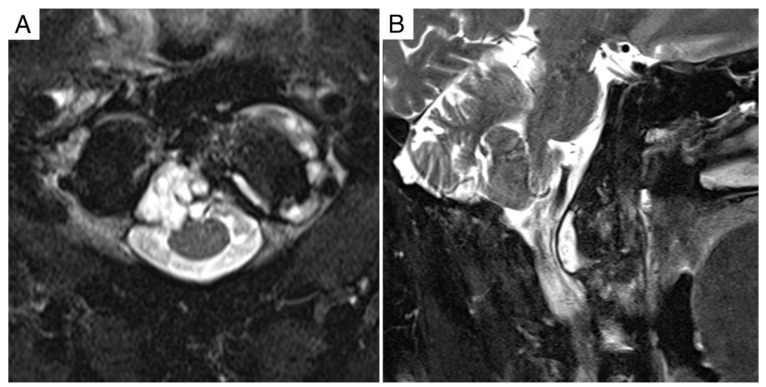
Case report 1: pre-operative MRI. (**A**) Axial view showing Bilski grade 1C epidural involvement. (**B**) Sagittal view showing extracompartimental extension.

**Figure 2 jcm-11-06058-f002:**
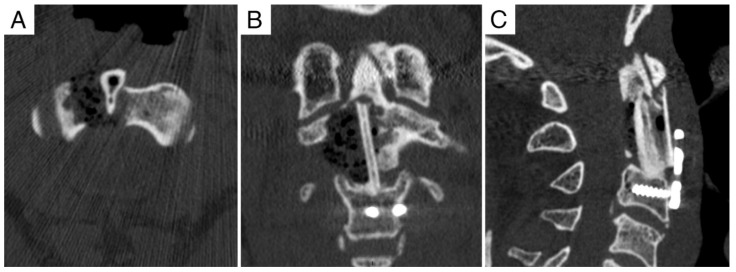
Case report 1: previous reconstruction of the anterior column with fibular graft and buttress plating. (**A**) Axial view. (**B**) Coronal view. (**C**) Sagittal view.

**Figure 3 jcm-11-06058-f003:**
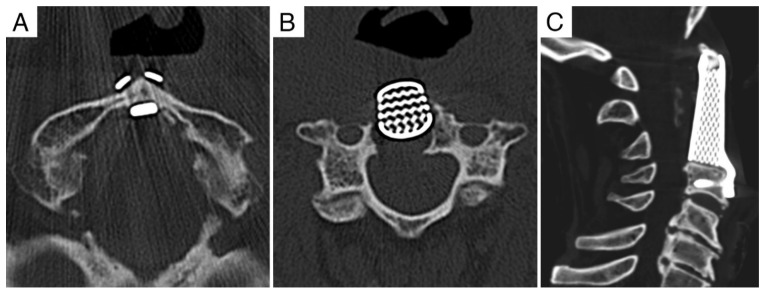
Case report 1: post-operative CT-scan. (**A**,**B**) Axial views, showing the prosthesis and its hooking to C1. (**C**) Sagittal view.

**Figure 4 jcm-11-06058-f004:**
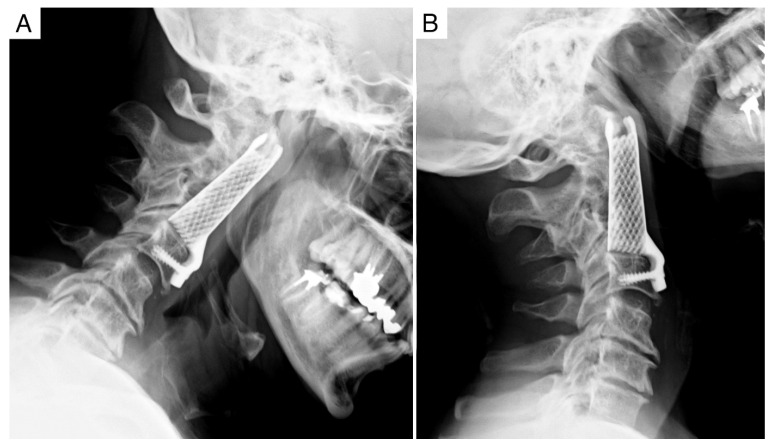
Case report 1: follow up at 9 months. (**A**,**B**) Dynamic X-rays showing no subsidence or displacement of the implant and stability of the spine.

**Figure 5 jcm-11-06058-f005:**
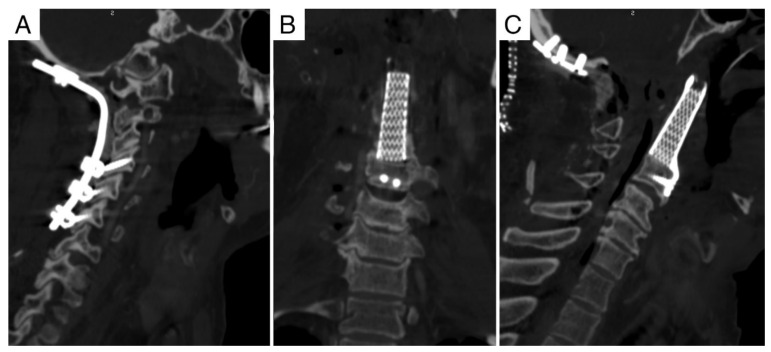
Case report 1: post-operative CT scan after revision. (**A**) Posterior instrumented C0-C6 fixation. (**B**,**C**) 3D-printed implant firmly osteo-integrated.

**Figure 6 jcm-11-06058-f006:**
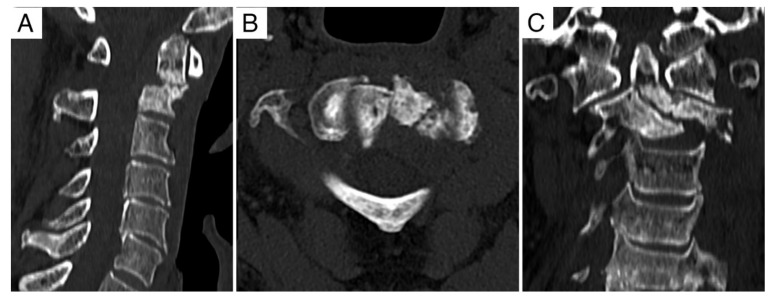
Case report 2: pre-operative CT. Pathological C2 fracture after surgery and radiation therapy for chordoma. (**A**) Sagittal view. (**B**) Axial view. (**C**) Coronal view.

**Figure 7 jcm-11-06058-f007:**
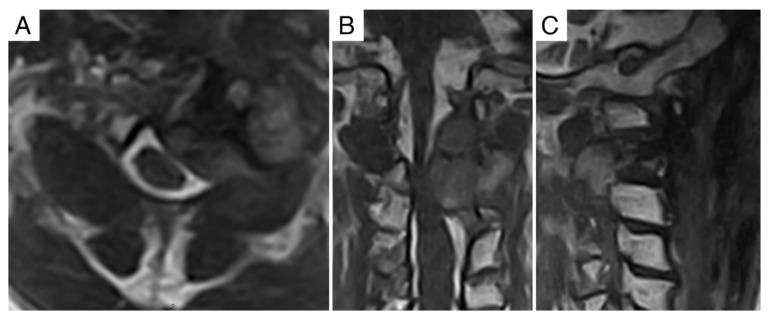
Case report 2: pre-operative MRI. (**A**) Axial view showing Bilski grade 2 epidural involvement (thus, causing spinal cord compression). (**B**) Coronal view. (**C**) Sagittal view.

**Figure 8 jcm-11-06058-f008:**
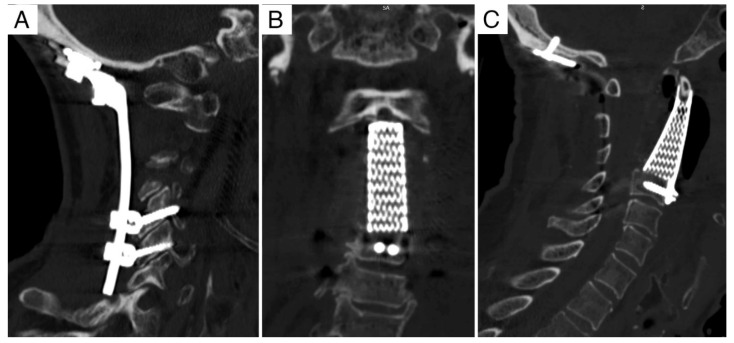
Case report 2: post-operative CT scan. (**A**) Posterior instrumented occipito–cervical fixation. (**B**,**C**) Anterior column reconstruction with 3D-printed custom made prothesis: distal fixation in C3 with integrated plate–screw system stabilized to the implant, proximal fixation obtained with a saddle that fitted the anterior arch of C1.

**Table 1 jcm-11-06058-t001:** Patients’ demographics.

Patient	Age (Years), Sex	Diagnosis	Enneking	WBB	Previous Treatments	Approach	Margin	Local Control (Months)	Overall Survival (Months)	Oncological Status
1	65, M	Recurrent chordoma C2	IB	8-5/A-D	RT (70.4 Gy carbon-ion, Δt = 4 y) Extracapsular debulking and reconstruction w/ allograft and carbon-fiber buttress plate (Δt = 2 y)	A	Intralesional	18	20	DOC
2	75, M	Recurrent chordoma C2	IB	8-3/A-D	Extracapsular debulking (Δt = 8 m) RT (74 Gy proton-ion, Δt = 2 m)	A+P	Intralesional	12	32	DOD

RT, radiation therapy; DOD, died of the disease; DOC, died of complications.

**Table 2 jcm-11-06058-t002:** Literature review.

Authors	Year	Article Type	*n*	Diagnosis	Age, Sex	Previous Treatments	Surgical Treatment	Approach		Adjuvant Treatments	Proximal Fixation	Distal Fixation	Follow-Up
Xu N et al. [9]	2016	Case report	1	C2 Ewing sarcoma	12, M	None	Single-level (C2) IL spondylectomy	P (C0-C6) + A (extraoral retropharyngeal) staged (2 weeks)		CHT + RT	Divergent TI-screws in C1 lateral masses	Titanium wiring in DVB (C3; TI-screws insertion not possible)	1 year
Li X et al. [10]	2017	Case report	1	C2-C3-C4 thyroid ca. metastasis	53, F	None	3-level (C2-C4) IL corpectomy + thyroidectomy and lateral lymph nodes dissection	A and P (C1-C6)		Radioiodine I^131^	Divergent TI-screws in C1 lateral masses	2 TI-screws in DVB (C5)	1 year
Mobbs RJ et al. [11]	2017	Case report	1	C1-C2 chordoma	63, M	CT-guided transoral biopsy	2-level (C1-C2) IL excision	P (C0-C3) and A (transoral)		RT	2 TiP-screws in the clivus	2 TI-screws in DVB (C3)	9 months
He S et al. [12]	2019	Case report	1	C2-C7 chondrosarcoma	27	Posterior open biopsy	6-level (C2-C7) IL en bloc resection	P (C0-T2) and A (retropharyngeal) staged (1 week)		Local cisplatinum intra-op.	Divergent TI-screws in C1 lateral masses	2 TiP-screws in DVB (T1)	14 months
He S et al. [13]	2019	Case series	7	C1-C3 (4 patients) C1-C2 (3 patients)	Average 47.6 ± 19.0 (range 12–72)	- - -	Multi-level IL spondylectomy (4 3-level, 3 2-level)	P and A (retropharyngeal) staged		- - -	TI-screws in occipital condyles	2 TiP-screws in DVB	Average 14.8 (range 7.3–24.2)
				- 5 chordoma - 1 Langerhans cell histiocytosis - 1 plasmocytoma									
Parr WHC et al. [14]	2020	Case report	1	C3-C4-C5 chordoma	45, M	CT-guided biopsy	3-level IL spondylectomy	P (C2-C6) and A staged (3 days)		RT (proton-beam)	2 TI-screws in C2	2 TI-screws in DVB (C6)	17 months
Li Y et al. [15]	2020	Case report	1	C1 plasmocytoma	57, M	CT-guided transoral biopsy	IL tumor debulking	A (retropharyngeal) and P (C0-C3)		RT (57 Gy/27 ft.)	1 TI-screw in occipital condyle	1 TI- and 1 TiP-screw in DVB (C2)	(Not reported)
Wei F et al. [16]	2020	Case series	9	7 C2 2 C2-C3	Average 31.4 (range 12–59)	CT-guided biopsy	Single- and multi-level IL spondylectomy	P (4 cases to C0, 5 to C1) and A (retropharyngeal) staged (average 14.4 days)		RT, CHT (in 2 cases)	Divergent TI-screws in C1 lateral masses	2 TiP-screws in DVB (not possible in 2 cases)	Average 28.6 (range 12–42)
				- 4 giant cell tumor - 2 chordoma - 1 Ewing sarcoma - 1 paraganglioma - 1 hemangioendothelioma									
Yang X et al. [17]	2020	Case report	1	C3-C4-C5-C6-C7-T1 chordoma	40, F	2 tumor debulkings	6-level IL extracapsular excision	A (retropharyngeal) and P (C1-T3)		None	3 TiP-screws in vertebral body of C2	3 TiP-screws in DVB (T2)	9 months
Hunn SAM et al. [18]	2020	Case report	2	C2 thyroid ca. metastasis	56, F	None	Single-level (C2) IL corpectomy	A (retropharyngeal)	and P (C1-C3)	RT and CHT	2 TiP-screws in C1 lateral masses	3 TiP-screws in DVB (C3)	14 months
				C2-C3 myeloma	63, M	None	2-level (C2-C3) IL corpectomy		and P (C1-C4)	(Not reported)		3 TiP-screws in DVB (C4)	4 months
Baldassarre BM et al. [19]	2021	Case report	1	C1-C2 chordoma	45	Biopsy	IL excision	P (C0-C4) and A (retropharyngeal) staged (1 day)		RT (proton beam)	Saddle for anterior arch of C1	2 TiP-screws in DVB (C3)	12 months

CHT, chemotherapy; RT, radiation therapy; IL, intralesional; TI, through-the-implant (0-profile); TiP, through integrated plate; DVB, distal vertebral body.

## Data Availability

Data supporting reported results can be found in the electronic medical records of the Institute.

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
