# Peer review of "Custom-Made 3D-Printed Implants for Anterior Column Reconstruction in the Upper Cervical Spine after Intralesional Extracapsular Excision—Report of 2 Cases and Literature Review"

_jcm, 2022, doi:10.3390/jcm11206058_

Round 1

Reviewer 1 Report

Chordomas remain rare malignant neoplasms, accounting for 1–4% of all primary bone tumors. Approximately 6% of chordomas are observed in the cervical spine. Dr. Girolami et al. presented interesting research pertaining to the usage of 3D printing in two cases of C2 chordoma. First of all, I would like to congratulate the authors on their choice, it is an indication of their good clinical experience and skills. Their paper may be helpful in similar cases of chordoma. In my opinion, the following issues should be addressed before publication:

Major:
- Title: the authors have focused on the small, but highly interesting part of the cervical spine. I suggest making the title more precise to highlight what the cases were about "C2 chordoma"

- The basic information about chordoma should be included in the introduction (e.g. based on WHO Classification of Tumours. 5th edition. Soft Tissue and Bone Tumours; https://www.ncbi.nlm.nih.gov/books/NBK430846/; 10.3390/jcm11144117).

- Materials and Methods: What is ~time of this procedure (from CT to final printed model)? Could you please add photos/screens from the following steps (even to Supplementary materials)?

- Line 137-138: the authors described these cases very well - in my opinion short table with the most important data (e.g. localization, type of lesion and follow-up) will be very helpful for readers. Was the 3d printing used in C2 chordoma cases previously only by Mobbs?

- The authors should elaborate more about alternative treatment options. What would be the best option in their cases if the 3d printed models wouldn't be available?

Minor:
line 23: typo in "3D-printed"
line 44-45: the proper reference should be added
line 51-52: please delete this extra space
line 137-138: the proper references should be added
line 158-9: please delete this extra space
line 162: please delete extra ")" and the extra "endter at the end of this sentence
overall comment: the current paper encompasses topics related to both orthopedics and neurosurgery. It may (should?) be stated that the role of 3D printing in these medical specialties is growing. 3D printing is used not only in reconstruction surgeries (e.g. 10.1017/cjn.2021.57, 10.3390/ma15144731), but also in surgery planning (e.g. 10.3340/jkns.2021.0235, 10.3390/jcm10061201) and in education (e.g. 10.1136/bmjstel-2017-000234).

Reviewer 2 Report

This is a very interesting and valuable study of 2 cases. While the introduction and other parts are detailed the materials and methods need to be more detailed for better perception of the case selection, design and manufacturing of the implant and surgery  

Author Response

We thank the reviewer for his appreciation of the manuscript. It has been re-edited and modified and we hope it will meet the reviewer’s expectations. A pdf document concerning the design and manufacturing of 3D-printed prosthesis has been added as Supplementary material.

Round 2

Reviewer 1 Report

Dear authors,I would like to congratulate you on your hard work. In my opinion, the paper can be published in its current form.